# Hiring of the Anti-Quorum Sensing Activities of Hypoglycemic Agent Linagliptin to Alleviate the *Pseudomonas aeruginosa* Pathogenesis

**DOI:** 10.3390/microorganisms10122455

**Published:** 2022-12-12

**Authors:** Maan T. Khayat, Tarek S. Ibrahim, Khaled M. Darwish, Ahdab N. Khayyat, Majed Alharbi, El-Sayed Khafagy, Mohamed A. M. Ali, Wael A. H. Hegazy, Hisham A. Abbas

**Affiliations:** 1Department of Pharmaceutical Chemistry, Faculty of Pharmacy, King Abdulaziz University, Jeddah 21589, Saudi Arabia; 2Department of Medicinal Chemistry, Faculty of Pharmacy, Suez Canal University, Ismailia 41522, Egypt; 3Department of Pharmaceutics, College of Pharmacy, Prince Sattam Bin Abdulaziz University, Al-Kharj 11942, Saudi Arabia; 4Department of Pharmaceutics and Industrial Pharmacy, Faculty of Pharmacy, Suez Canal University, Ismailia 41552, Egypt; 5Department of Biology, College of Science, Imam Mohammad Ibn Saud Islamic University, Riyadh 11432, Saudi Arabia; 6Department of Biochemistry, Faculty of Science, Ain Shams University, Abbassia, Cairo 11566, Egypt; 7Department of Microbiology and Immunology, Faculty of Pharmacy, Zagazig University, Zagazig 44519, Egypt; 8Department of Pharmaceutical Sciences, Pharmacy Program, Oman College of Health Sciences, Muscat 113, Oman

**Keywords:** *Pseudomonas aeruginosa*, linagliptin, quorum sensing, antimicrobial resistance, bacterial virulence, drug repurposing

## Abstract

Bacteria communicate with each other using quorum sensing (QS) which works in an inducer/receptor manner. QS plays the main role in orchestrating diverse bacterial virulence factors. *Pseudomonas aeruginosa* is one of the most clinically important bacterial pathogens that can cause infection in almost all body tissues. Besides its efficient capability to develop resistance to different antibiotics, *P. aeruginosa* acquires a huge arsenal of virulence factors that are controlled mainly by QS. Challenging QS with FDA-approved drugs and natural products was proposed as a promising approach to mitigate bacterial virulence enabling the host immunity to complete the eradication of bacterial infection. The present study aims to evaluate the dipeptidase inhibitor-4 inhibitor hypoglycemic linagliptin anti-QS and anti-virulence activities against *P. aeruginosa* in vitro, in vivo, and in silico. The current results revealed the significant ability to diminish the production of protease and pyocyanin, motility, and biofilm formation in *P. aeruginosa*. Furthermore, the histopathological examination of liver and kidney tissues of mice injected with linagliptin-treated bacteria showed an obvious reduction of pathogenesis. Linagliptin downregulation to QS-encoding genes, besides the virtual ability to interact with QS receptors, indicates its anti-QS activities. In conclusion, linagliptin is a promising anti-virulence and anti-QS candidate that can be used solely or in combination with traditional antimicrobial agents in the treatment of *P. aeruginosa* aggressive infections.

## 1. Introduction

*Pseudomonas aeruginosa* is a pervasive nosocomial pathogen that can infect almost all tissues causing a wide range of acute and chronic infections, such as surgical and burn wounds, eye, lung, bloodstream, urinary tract, and central nervous system infections [1,2,3,4]. *P. aeruginosa* is listed among the most serious pathogens: for instance, it was the second most frequent pathogen that was isolated from patients admitted to intensive care units [5,6]. The *P. aeruginosa* pathogenesis is worth contemplating because of its horrible ability to invade and infect host cells, which is owed to a huge arsenal of virulence factors that are regulated in a magnificent highly organized manner [7,8]. Structurally, *P. aeruginosa* recruits efficient locomotion flagella and adhesion pili organs, which in addition to its small size facilitates its spread and invasion into the host tissues [7,9]. Furthermore, *P. aeruginosa* can survive in very low nutrient concentrations and resist antimicrobial agents, which could explain its high prevalence in serious nosocomial infections [7,10,11]. *P. aeruginosa* pathogenesis is guaranteed by its ability to destroy diverse extracellular enzymes such as proteases, elastases, hemolysis, nucleases, and others, besides the formation of biofilm that establishes its infection in almost all tissues and defeats the host immunity [2,3,6,12]. Additionally, *P. aeruginosa* has an exceptional capacity to develop heritable resistance to several antibiotics, which necessitated the innovation of specific anti-pseudomonal antibiotics; however, *P. aeruginosa* developed resistance to these agents [8,13,14,15]. This intricate situation makes *P. aeruginosa* one of the major pathogens that have to be defeated either by developing new antibiotics or by the invention of new treatment strategies and approaches [7,16].

*P. aeruginosa* employs different systems to regulate its pathogenesis; the quorum sensing (QS) system play a key role in the regulation of the *P. aeruginosa* arsenal of virulence factors [10,17,18]. The QS system regulates the production of invasive extracellular enzymes, biofilm formation, motilities, and production of virulence factors such as pyocyanin and others [1,19,20]. The QS system is the communicative system that bacteria use to arrange their pathogenesis in the population, where they produce autoinducers (AIs) in the surrounding niche to bind into their cognate receptors (QS receptors). The formed AIs-QS receptor binds to specific sequences on the bacterial chromosome to regulate the expression of virulence genes [21,22,23]. As in almost Gram-negative bacteria, *P. aeruginosa* senses a diverse range of n-acetyl homoserine lactone (AHLs) autoinducers on basically three main QS receptors [10,13,20,24,25,26]. Bearing in mind the above facts, targeting the *P. aeruginosa* QS system could efficiently curtail its virulence and diminish its pathogenesis. 

The development of bacterial resistance constitutes a considerable global health challenge, in particular in the light of the decreased supply of new effective antibiotics. This dictates the need to find new efficient clinical regimens and the development of new approaches to overcome bacterial resistance [27,28,29,30,31]. Targeting bacterial resistance is an interesting approach to counteract the development of bacterial resistance that confers several advantages. First, the attenuation of bacteria facilitates bacterial killing by the immune system without influencing bacterial growth, which urges bacteria to develop resistance [23,24,32]. Moreover, this approach does not affect bacterial flora [23,32]. In this direction, numerous chemical moieties were screened for their anti-virulence activities [33,34] and attention was drawn to safe compounds of natural origin or approved safe drugs [3,19,21,22,31,35]. 

In particular, the repurposing of already approved safe drugs as anti-virulence agents has additional merits, such as saving costs and time needed to perform extended pharmacological, toxicological, and pharmaceutical studies [2,19,21,22,35,36,37]. In previous leading studies, a dipeptidyl peptidase-4 inhibitor (DPI-4) antidiabetic sitagliptin that harbors pyrazine dicarboxylic acid and triazole chemical moieties showed a significant ability to cripple the *Serratia marcescens* [19,20] and *P. aeruginosa* [2,38]. Sitagliptin significantly showed anti-QS activity and anti-virulence activities in vitro and in vivo. That encourages us to extend our screening of anti-virulence candidates among other gliptins that share similar chemical moieties with sitagliptin. The current study was designed to assess the anti-virulence and anti-QS activities of linagliptin in vitro and in vivo. Furthermore, the effects of linagliptin on the expression of QS-encoding genes were evaluated and a molecular in silico study was conducted to attest to the interference ability of linagliptin with QS receptors. This study is a preliminary study intended to evaluate the possibility of linagliptin employment to serve as an anti-virulence agent alone or in addition to traditional antibiotics.

## 2. Materials and Methods

### 2.1. Microbiological Media, Chemicals, and Bacterial Strain

All the microbiological media used were purchased from Oxoid (Hampshire, UK). Linagliptin (Cas No. 668270-12-0) was obtained from Cayman Chemicals (Ann Arbor, MI, USA). All the chemicals used were of analytical grade. *P. aeruginosa* PAO1 was employed as a bacterial model to evaluate the linagliptin anti-virulence effect.

### 2.2. Detection of MIC of Linagliptin

The linagliptin MIC was determined against *P. aeruginosa* PAO1 by the broth microdilution method according to the Clinical Laboratory and Standards Institute Guidelines (CLSI, 2015) [6,25]. 

### 2.3. Evaluation of the Effect of the Linagliptin on P. aeruginosa Growth

To emphasize that the linagliptin anti-virulence activity was not due to its inhibition of bacterial growth, the linagliptin anti-virulence activities were evaluated at sub-MIC (1/5 MIC). Furthermore, the effect of linagliptin at sub-MIC (1/5 MIC) on bacterial growth was evaluated as described earlier [2,19]. The bacterial cells were viably counted in the presence or absence of linagliptin at sub-MIC [3,6]. In brief, after culturing bacteria with or without linagliptin at sub-MIC for 24 h at 37 °C; the bacterial cells were serially diluted and 100 µL from each dilution was spread on the surface of Muller Hinton agar plates. After overnight incubation, the bacterial cells were counted and calculated as CFU/mL. 

### 2.4. Determination of the Antibiofilm Activity of Linagliptin 

The crystal violet method was used as described previously [6,26,39] to assess the inhibitory effect of linagliptin on biofilm formation. Briefly, 100 µL aliquots of *P. aeruginosa* adjusted to 0.5 MacFarland optical density were mixed or not with 100 µL of LB broth provided with linagliptin to obtain sub-MIC concentration into microtiter plates. After 24 h incubation at 37 °C, the non-biofilm-forming bacterial cells were washed out and the adhered biofilm-forming cells were stained with crystal violet 1% for 25 min. The adhered crystal violet was extracted with 33% glacial acetic acid and the absorbance was measured at 590 nm. 

The formed biofilms were visualized as described earlier [21,22,34]. The bacterial biofilms were allowed to form on coverslips in the presence or absence of linagliptin at sub-MIC. The coverslips were stained with crystal violet and examined under the light microscope.

### 2.5. Evaluation of the Effect of Linagliptin on P. aeruginosa Motility 

To evaluate the inhibitory effect of linagliptin on *P. aeruginosa* swarming motility, LB agar plates containing linagliptin at sub-MIC were prepared. The LB plates were centrally inoculated with 5 µL of optically adjusted fresh overnight *P. aeruginosa*. Control plates were prepared and centrally inoculated with *P. aeruginosa*. The swarming zones were measured in mm [2,20,40]. 

### 2.6. Determination of the Effect of Linagliptin on the Protease Production

The skim milk agar method was employed as previously described [2,37,40] to evaluate the linagliptin inhibitory effect on the activity of protease at its sub-MIC. The supernatants containing extracellular enzymes from *P. aeruginosa* overnight cultures provided or not with linagliptin at sub-MIC were collected by centrifugation. In 5% skim milk agar plates, 100 μL of collected supernatants were transferred into prepared wells and incubated for 24 h at 37 °C. The proteolytic activity was observed as clear zones around wells; clear zones are measured in mm. 

### 2.7. Detection of the Effect of Linagliptin on Pyocyanin Production

The production of pyocyanin was assayed in the presence or absence of linagliptin as previously described [2,3,34]. In Eppendorf tubes, the overnight fresh culture of *P. aeruginosa* (10 μL) was mixed with 1 LB broth provided or not with linagliptin at sub-MIC. The Eppendorf tubes were centrifuged after 48 h incubation at 37 °C, and the absorbances of diffusible pigment were measured at 691 nm.

### 2.8. Quantification of QS-Encoding Genes

To assess the effect of linagliptin at sub-MIC on the expression of *P. aeruginosa* QS-encoding genes, quantitative real-time PCR was used. First, the RNA was extracted from *P. aeruginosa* treated or not with linagliptin at sub-MIC using a Purification Kit Gene JET RNA (Thermoscientific, Waltham, MA, USA) as described [41,42,43]. The RNA samples were stored until needed at −80 °C. The expression levels of the tested genes were calculated using the comparative threshold cycle (∆∆Ct) method, as described [2,22,41]. The expression levels were standardized to the expression of the housekeeping gene *ropD*. To synthesis the cDNA, a high-capacity cDNA reverse transcriptase kit (Applied Biosystem, Waltham, MA, USA) was employed, and a Syber Green I PCR Master Kit (Fermentas, Waltham, MA, USA) was used for amplification in a Step One instrument (Applied Biosystem, Waltham, MA, USA). Primers used in this study are listed in Table 1.

### 2.9. Histopathological Evaluation

To evaluate the in vivo anti-virulence activity of linagliptin, mice were injected with linagliptin at sub-MIC, and histopathological examination was performed on kidney and liver tissues as previously described [44,45]. Three-week-old *Mus musculus* mice (albino mice) were arranged in five groups, each comprising 5 mice. The test group was intraperitoneally injected with *P. aeruginosa* (1 × 10^6^ CFU/mL) treated with linagliptin at sub-MIC. The second and third mice groups were intraperitoneally injected with untreated *P. aeruginosa* (1 × 10^6^ CFU/mL) or *P. aeruginosa* treated with DMSO as positive control groups. The fourth and fifth groups were injected with sterile PBS or kept un-injected as negative control groups. After five days of observation, the mice were euthanized by cervical dislocation, and livers and kidneys were dissected and rinsed with normal saline. The tissues were fixed in neutral buffered formalin (10%). For histopathological examination, they were dehydrated in increasing concentrations of ethanol (70%, then 90%, then 100%). Then, the sample tissues were cleared in xylol, steeped, and embedded in paraffin wax. Rotatory microtome was used to cut sections in 5 μm thickness that were stained with hematoxylin and eosin (H&E) for light microscope 

### 2.10. Molecular Docking Study

All ligands were constructed with MOE-2019.01 suite (Quebec, QC, Canada) using its isomeric/canonical SMILES strings. Ligands were energy minimized under MMFF-94S modified force field as previously described [46,47,48]. Biological targets LasR (PDB; 6MVN), QscR (PDB; 6CC0) and LasI (PDB; 1RO5), were prepared using MOE-2019.01 3D protonation (300 K, pH = 7.4 0.1 M salt solution under GB-VI implicit solvent model), autocorrected for atoms orders, bond connectivities, and partial charge. Binding sites were defined with the MOE-Alpha Site Finder geometrical protocol for including important pocket amino acids described within the current literature and thoroughly described in the presented manuscript. Induced-fitting docking protocol was adopted allowing relevant flexibility for the pocket’s amino acids. Ligand conformations were generated on the fly through ligand placement/bond rotations inside the defined active pocket and guided by the triangular-matching technique [49]. Obtained ligand conformations were ranked using LondondG and the top ten docked binding modes were subjected to refinement and energy minimization where the protein’s residue sidechains were tethered. Rescoring was done for the refined binding modes via the GBSVI/WSA-dG force field depending on current loading partial charges, explicit solvation electrostatic, exposure-dependent surface area, as well as ligand–protein Coulombic’s electrostatic and Lenard-Jones van der Waals scoring values [50,51]. Selection of the best ligand docking binding mode was done considering high negative docked energy scores, low RMSDs ≤ 2 Å cutoffs, and/or relevant strong interactions with important pocket amino acids reported within the literature. PyMol2.0.6 (Schrödinger, NY, USA) and MOE wizard/measurement tools were used for visualization [52].

### 2.11. Statistical Analysis

All the experiments were conducted in triplicate, and are presented as mean ± SD. Unpaired *t*-tests and Graph Pad Prism 8 were used to examine the significance of the linagliptin inhibitory activities against bacterial motilities. Paired *t*-tests and Graph Pad Prism 8 were employed to attest the significance of the linagliptin effects on growth, biofilm formation, protease, and pyocyanin. One way-ANOVA test followed by the Dunnett post-test was used to attest to the significance of downregulation of QS genes. *p* values < 0.05 were considered statistically significant

## 3. Results

### 3.1. Effect of Linagliptin on the P. aeruginosa Growth

The minimum inhibitory concentration (MIC) of linagliptin against *P. aeruginosa* was determined using the broth microdilution method. The least concentration of linagliptin that inhibited the *P. aeruginosa* growth was 10 mg/mL. To exclude any effect of linagliptin on *P. aeruginosa* growth, *P. aeruginosa* was overnight grown in the presence or absence of linagliptin at sub-MIC (1/5 MIC 2 mg/mL), and the optical densities of bacterial cultures were measured (Figure 1). There was no significant difference between the bacterial growth in the presence or absence of linagliptin at sub-MIC. For further confirmation, the bacterial cells were counted, and there was no significant difference between bacterial counts in the presence or absence of linagliptin at sub-MIC. It is important to note that all the next experiments were performed using linagliptin at sub-MIC (2 mg/mL). 

### 3.2. Antibiofilm Activity of Linagliptin 

The crystal violet method was used to investigate the linagliptin antibiofilm activity at sub-MIC (2 mg/mL). The absorbances of the crystal violet-stained *P. aeruginosa* biofilm-forming cells treated with linagliptin at sub-MIC were measured and presented as percentage change from untreated bacterial cells (Figure 2). Linagliptin significantly inhibited the formation of biofilms by *P. aeruginosa*. Furthermore, the formed biofilms on coverslips were stained with crystal violet and examined under a light microscope. Obviously, linagliptin reduced the biofilm formation.

### 3.3. Linagliptin Cripples the P. aeruginosa Motility

LB agar plates were prepared with or without linagliptin at sub-MIC and centrally inoculated with *P. aeruginosa*. The bacterial swarming zones were measured (Figure 3). Linagliptin significantly diminished the *P. aeruginosa* motility.

### 3.4. Linagliptin Decreases the Production of Protease

The linagliptin’s ability to reduce the production of protease was assessed by employing the skim agar plate method. The extracellular produced protease was collected from *P. aeruginosa* cultures grown in the presence or absence of linagliptin at sub-MIC. The formed clear zones around the wells representing the proteolytic activity were measured (Figure 4). Linagliptin significantly reduced the production of protease. 

### 3.5. Linagliptin Decreases the Production of Pyocyanin

Fresh *P. aeruginosa* inoculums were allowed to produce its bluish-green virulent pigment on LB broth provided or not with linagliptin at sub-MIC. The absorbances of the produced pigment were measured (Figure 5). Linagliptin significantly reduced the production of pyocyanin. The data are presented as percentage change from untreated control. 

### 3.6. Linagliptin Diminishes the P. aeruginosa Pathogenesis 

To depict the linagliptin’s effect on diminishing *P. aeruginosa*-induced pathogenesis, representative photomicrographs were captured for renal and liver tissues of infected mice with *P. aeruginosa* treated or not with linagliptin at sub-MIC. In negative control groups of mice which were kept un-infected or injected with sterile PBS, the liver and kidney tissues showed normal cellular details and tissue architectures (Figure 6A,G). In the positive control mice injected with untreated *P. aeruginosa*, the blood vessels of liver tissues were severely congested, associated with perivascular fibrosis and hydropic degeneration of hepatocytes (Figure 6B,C). Furthermore, degenerative changes, swelling, and areas of cellular proliferation were observed in renal tubules and caseous necrosis in kidney tissues of the mice injected with untreated *P. aeruginosa* (Figure 6H,I). On the other hand, the liver tissues of mice injected with *P. aeruginosa* treated with linagliptin showed mild infiltration of von Kupffer cells, vacuolation of a few hepatocytes, and mild congestion in hepatic blood vessels (Figure 6D–F). Additionally, linagliptin diminished the *P. aeruginosa* pathogenesis in kidney tissues, where mild diffuse cystic dilation of renal tubules, fewer focal areas of cellular infiltration, and normal renal cortex were observed (Figure 6J–L). These findings indicate the relieving effect of linagliptin at sub-MIC on *P. aeruginosa* pathogenesis. 

### 3.7. Linagliptin Anti-Virulence Activity Is Owed to Interfering with QS Systems

The QS system regulates the *P. aeruginosa* pathogenesis and orchestrates the expression of several virulence factors. QS systems in *P. aeruginosa* regulate biofilm formation, bacterial motility, and production of diverse extracellular enzymes, and the virulent pigment pyocyanin [3,6]. *P. aeruginosa* employs mainly three QS systems; two are Lux homologs, namely LasI/R and RhlI/R, and one is a non-Lux-type Pqs system [2,3,6]. In addition to the orphan homolog of LuxR, QscR does not have a partner LuxI but binds to the LasI autoinducers [2,6]. 

#### 3.7.1. Linagliptin Downregulates the Expression of the Encoding Genes of Three *P. aeruginosa* QS Systems

To assess the linagliptin effect at sub-MIC on the expression of QS-encoding genes in *P. aeruginosa*, qRT-PCR was performed (Figure 7). The levels of QS-encoding genes were assessed in linagliptin-treated and untreated *P. aeruginosa* employing the 2^−∆∆Ct^ method. The expression levels of autoinducer synthetases encoding genes *lasI*, *rhlI*, and *pqsA* and their cognate receptors encoding genes *lasR*, *rhlR*, and *pqsR*, respectively, were significantly reduced in linagliptin-treated *P. aeruginosa* samples in comparison to untreated control. 

#### 3.7.2. Multi-Target Docking Analysis of Linagliptin on *P. aeruginosa* QS Systems

A validated molecular docking workflow was conducted for explaining the molecular bases of linagliptin’s (LIN; Figure 8) in vitro anti-pseudomonal activity. The *P. aeruginosa* LasI acyl-homoserine lactone (HSL) synthetase, as well as LasR and QscR quorum-sensing (QS) systems, were adopted. The LasI protein is composed of a triple layer α/β/α sandwiched platform comprising of a V-shaped binding site represented with deep elongated tunnel/cleft allowing substrate fitting (Figure 9A) [53]. The LasR and QscR QS proteins are deposited as cyclic protein homodimers solved with their agonist pheromones, 3-O-C10-HSL and C12-HSL, respectively (Figure 9B,C) [54,55]. Both LasR and QscR are with amino-terminal α/β/α sandwiched ligand binding domain whereas the DNA-binding site is at their respective carboxy terminus. Binding sites are generally elongated across the target structures. In all three QSs, the binding ligands are considered fully embedded inside the target pockets, permitting them to mediate several non-polar interactions plus a few hydrogen bondings. The co-crystallized ligands directed their respective lactone heads towards the small inner hydrophilic sub-pocket, stabilizing these heads via hydrogen and hydrophobic interactions. The ligands’ amide portions are anchored at close proximity to the pocket’s polar residues, allowing hydrogen bond interactions with corresponding H-bond acceptor/donor moieties of the pocket’s amino acids. Finally, the non-polar acyl tails of the co-crystallized ligands are inserted deeply into a larger hydrophobic sub-pocket lined with a few polar amino acids.

##### Docking Analysis at LasI-Type AHL Synthase Binding Site

The investigated ligand showed favored orientation within the LasI’s elongated cleft as well as a V-shaped sub-pocket. The binding mode of LIN was quite interesting where the ligand’s 3-aminopiperidine group was docked at the V-shaped cleft, which has been considered sterically favored (Figure 10A). Such orientation allowed the rigid butynyl substitution, on the core xanthine ring, to be directed towards the open/solvent side of the V-shaped cleft for minimized steric hindrance with pocket-lining amino acids. 

The stability of this binding mode is mediated by double hydrogen bond pairing between LIN’s free amine and Arg30 sidechain as well as hydrophobic contact across LIN’s piperidine hydrocarbon cage and Phe27/Trp33 sidechains (Appendix A). The stability of LIN at the elongated tunnel was further achieved through hydrophobic contacts with non-polar residues of the pocket’s middle and terminal sides as well as polar interactions with Thr124 and Thr144 sidechains. Based on the above docking findings, LIN was denoted with a high negative dock score (*S* = −7.1987 Kcal·mol^−1^) conferring its preferential binding towards the LasI binding site. 

Since the PDB protein was deposited without a native co-crystallized ligand, it was highly recommended to proceed through docking validation for the above-described *P. aeruginosa* LasI in silico investigation. Validation was done by investigating the molecular docking simulation of the LasI’s reported reference antagonist, TZD-C8. To our delight, TZD-C8 showed significant adherence to its reported binding mode as well as comparable orientation to that of the investigated anti-pseudomonal gliptin. The reference control ligand was first introduced as a synthetic blocker of *P. aeruginosa* LasI QS synthetase exhibiting an altered QS signaling pathway, which hampered swarming motility, and showed strong biofilm formation inhibition [56].

Here, in the presented docking study and that previously reported, the TZD-C8 polar thiazolidinedione head depicted strong hydrogen bonding with Arg30 and Ile107, ensuring their importance for ligand–target complex stabilization despite the ligands’ great structural diversity (Figure 10B). The latter residue-wise ligand–target stability interaction was confirmed through reported mutagenesis for both residues, showing abolished TZD-C8/LasI inhibition activity after double site-directed mutagenesis (I107S and R30D). The elongated LasI tunnel site harbored the TZD-C8′s non-polar tail where its inherited flexibility allowed successful maneuvers that would minimize any potential steric clashes with pocket-lining amino acids. Interestingly, TZD-C8 depicted a moderate docking score (*S* = −5.5102 Kcal·mol^−1^), being inferior to the tested gliptin. Further validation was ensured since the docked and re-docked TZD-C8 ligand exhibited low root-mean-square deviation (RMSD = 1.0712 Å) [57,58,59]. Typically, showing RMSD < 2 Å confers that the adopted docking algorithms and parameters were appropriate to determine the best binding modes and so in turn, the docking energies [60].

##### Docking Analysis at QscR-Type Quorum-Sensing Protein Binding Site

Further exploration of LIN’s pseudomonal biofilm inhibition activity was achieved through in silico prediction of the drug’s molecular aspects of QscR QS binding, since this target regulates the transcriptions of several bacterial virulence genes. The ligand illustrated favorable accommodating owing to relevant interactions with important QscR pocket residues. The docking study was also validated through a triple approach, where firstly the re-docked co-crystallized autoinducer (C12-HSL) adopted showed low RMSD (1.0850 Å) at QscR binding site (Appendix A).

Since the QscR co-crystallized ligand is an autoinducer with reported low nanomolar activity (EC_50_~5.0 nM), a second docking approach was to use a reported antagonist control reference ligand to evaluate the LIN’s antagonistic potentiality. Literature reports for a synthetic phenyl-HSL analog, known as Q9, showed weak *P. aeruginosa* QscR activating profiles with EC_50_ more than 70 nM [54] as well as an 80% reporter activation inhibited with inhibition activities at the mid-nanomolar range (IC_50_~26 nM) being represented as the most potent known QscR inhibitor [61].

Interestingly, comparable orientation and conformation were illustrated for both LIN and Q9 reference antagonist (Figure 11A). LIN’s terminal aromatic feature was anchored at the QscR’s large-sized non-polar sub-pocket, offering minimal steric clashes for such a huge scaffold. On the other hand, the ligand’s more polar fused ring (3-amino-piperidine moiety) was perfectly docked within the QscR’s small sub-pocket. However, a quite more extended conformation for LIN was depicted as compared to the same ligand within the LasI’s active site. Nevertheless, LIN’s conformation within the QscR binding site was quite tight owing to its highly steric inflexible butynyl group. The calculated Richard’s solvent-accessible surface area (SASA)/volume was higher for QscR as compared to LasI’s (579.64 Å^2^/331.18 Å^3^ versus 496.61 Å^2^/254.06 Å^3^). Concerning the depicted LIN’s orientation/conformation, the ligand’s ionizable head showed great superimposition with the C12-HSL crystallized lactone ring as well as that of QscR’s synthetic antagonist, Q9. On the other side, the ligand’s respective hydrophobic/aromatic moieties were directed into the large lipophilic QscR sub-pocket (Figure 11B). 

Several pocket amino acids were assigned important for ligand anchoring. Polar interactions with the anionic Asp75 sidechain were shown for docked LIN (Figure 11A). A sidechain of Asp75 was suggested as significant for LIN anchoring at QscR’s small sub-site since the earlier showed proximity towards the ligand’s NH or poor electronegative (δ^+^)/tautomeric nitrogen. Further stabilization of the docked LIN was achieved with wide-range polar amino acids including Ser38, Tyr66, and Ser129 (Appendix A).

It is worth mentioning that Ser38 and/or its vicinal residue, Tyr66, illustrated significant hydrogen bond pairing with LIN. Besides the ligand’s polar contacts, LIN also showed relevant van der Waals hydrophobic contacts with Phe39, Ala41, Tyr52, Tyr58, Trp62, Ile77, Leu82, Phe101, Trp102, Pro117, Ile125, and/or Met127 amino acids. Additionally, π-ring stacking with Phe54 as well as π-H interactions with Phe39, Tyr52, Phe54, Tyr58, Tyr66, Trp90, and/or Trp102 were also depicted (Appendix A). Non-polar van der Waals binding with Arg42 sidechain hydrocarbons was also depicted, all of which added to the significant docking score of LIN (*S* = −7.9967 Kcal·mol^−1^).

A comparable residue-wise binding profile was shown for Q9, where polar contacts with sidechains of Ser38, Tyr58, and Asp75 were depicted for the ligand’s amidic lactone head (Figure 11B). Moreover, relevant non-polar contacts with Phe54, Arg42, Tyr52, Tyr58, Tyr66, Ile77, Leu82, Pro117, Ile125, and Met127 were also depicted via Q9′s aromatic/aliphatic lipophilic tail, summing up a high-negative docking score (*S* = −8.2341 Kcal·mol^−1^). Validation of the Q9 binding pose was confirmed when being assigned a very low RMSD of 0.8445 Å in relation to the co-crystallized ligand, since great superimposition of the ligands’ lactone ring and amidic acyl groups were depicted.

##### Docking Analysis at LasR-Type Quorum-Sensing Protein Binding Site

A comprehensive exploration of LIN’s biofilm inhibition activity further proceeded through molecular docking simulation at another QS protein, LasR. Since the deposited LasR is co-crystallized with its autoinducer (EC_50_~15 nM), the docking simulation study was also validated using a reported positive control LasR antagonist. The same phenyl derivative HSL analog, Q9, that has been used for the QscR study illustrated an additional 10-fold more inhibitory activity than any of LasR’s best inhibitors. Moreover, this Q9 synthetic inhibitor has the advantage of lacking any atypical partial agonist characteristics, which are usually depicted with other LasR antagonists when being used at high concentration levels [61]. Further docking protocol validation was confirmed via redocking the co-crystallize ligand where a significantly low RMSD value (1.0576 Å) was obtained, ensuring the biological significance of the obtained docking binding modes and so the energies (Appendix A).

The docked LIN-LasR complex illustrated comparable ligand orientations with their respective at the QscR transcriptional protein. The ligands’ polar heads were perfectly anchored at the small sub-pocket, having its nitrogen at significant superimposition with the amide groups of both the co-crystalline and reference ligands (Figure 12). On the other hand, the LIN’s terminal aromatic scaffold was perfectly anchored at LasR’s large hydrophobic site, similar to the Q9′s terminal acyl functionality. Although having comparable orientation, LIN within LasR illustrated different conformation in relation to the same ligand at QscR. Typically, LIN adopted elongated almost straight conformation at QscR with its terminal scaffold being extended at both far ends of the target binding site. Nevertheless, the same ligand at LasR exhibited a crescent-like conformation with its terminal aromatic moiety being directed to the opposing face of the binding pocket. Such conformation could be reasoned for the comparatively higher QscR binding site SASA/volume (579.64 Å^2^/331.18 Å^3^) as compared to those of LasR (258.43 Å^2^/120.24 Å^3^). 

The polar interaction with the anionic amino acid, Asp73, was depicted owing to close proximity of its respective hydrogen bond pair towards the Asp73 oxygen atom. Additional strong hydrogen bonding with the Ser129 sidechain was also seen being mediated via the hydrogen bond acceptor of the ligand’s respective scaffold (Appendix A). Additional hydrogen bond pairings were predicted for Trp60 sidechain *ε N*-atom, all of which assign LIN a moderate-to-high negative docking score (*S* = −6.3114 Kcal·mol^−1^) (Figure 12A). A comparable polar interaction pattern was assigned for the Q9 control antagonist (*S* = −8.3188 Kcal·mol^−1^), having its amidic linker and lactone scaffold with favored strong hydrophilic interactions with Tyr56, Asp73, Tyr93, and Ser129 amino acids (Figure 12B). 

Concerning the ligand accommodation at the large hydrophobic site, ligands’ non-polar scaffolds were significant for mediating extra stability at LasI’s target pocket. Comparable hydrophobic interactions were assigned for LIN and reference ligands with Ile52, Trp60, Tyr93, Phe101, Ala105, Leu110, and Ala127. Further LasI-ligand complex stability was mediated with relevant π-driven and van der Waal interactions with LasI’s terminal end pocket non-polar residues. Both docked ligands exhibited close-range contacts with Leu40 and Leu125 as well as π-π and/or π-H interactions via their terminal non-polar moieties (Appendix A).

## 4. Discussion

The success of antibiotics since their first use in the 1940s in controlling the epidemics of aggressive bacterial infections is considered one of the greatest public health achievements. However, resistance development diminishes this great success [62,63]. Aiding the cause of resistance development is the misuse and/or abuse of antibiotics, which allow the bacteria to develop resistance [62]. As a consequence, the surviving bacterial cells transfer the resistance traits to new generations or even share them with other bacteria from the same or different genera [62,63]. Unfortunately, the development of resistance is observed in almost all classes of antibiotics that are harbored chromosomally or extra-chromosomally [36,64]. This failure in overcoming bacterial resistance worsens the patient’s infection and burdens their immunity [42,65]. This situation forced scientists and clinicians to consider new approaches and strategies. Among the approaches whose efficacy has been proven is the targeting of bacterial virulence [20,35]. This approach guarantees the mitigation of bacterial virulence, leaving the completion of their eradication to immunity without stressing bacterial cells to develop resistance [3,21,22,66]. In this direction, several natural ingredients and safe drugs have been repurposed to be used as adjuvants to antibiotics in the treatment of resistant infections [24,35,40,67]. 

*P. aeruginosa* is a clinically important nosocomial pathogen as it causes severe infections, for instance eye, wound, burn, and respiratory infections [9,11,68]. Most importantly, *P. aeruginosa* magnificently managed to develop resistance not only to antibiotics but to disinfectants and antiseptics [9,68]. This great pathogenesis in addition to resistance to antimicrobials ranks *P. aeruginosa* among the most serious bacterial pathogens that are listed in the ESKAPE list [3,6]. The current study aimed to investigate the anti-virulence activities of DPI-4 antidiabetic linagliptin against *P. aeruginosa*.

The effective implementation of effective anti-virulence and anti-QS agents for treating bacterial infections basically requires that the used agents do not affect bacterial growth [22,69]. To avoid the effect of linagliptin on *P. aeruginosa* growth, the anti-virulence activities of linagliptin were assayed at its sub-MIC concentration. For more emphasis, the optical densities and bacterial counts were performed for *P. aeruginosa* cultures in broth provided or not with linagliptin at sub-MIC, and there was no significant difference between bacterial growth in the absence or presence of linagliptin at sub-MIC. The *P. aeruginosa*’s ability to form biofilms is an additional hazard that explains the aggressive nosocomial infections and high resistance to disinfectants [28]. The antibiofilm agents increase the susceptibility of bacterial cells to antibiotics in vitro and in vivo [32,37]. The current findings showed a significant ability of linagliptin to diminish biofilm formation. The bacterial motility ensures more spreading of bacterial infections: it was observed that the motile bacteria were more virulent than non-motile mutants [70]. Furthermore, curtailing bacterial motility mitigates their virulence [17,23]. Interestingly, linagliptin significantly crippled *P. aeruginosa* motility. 

*P. aeruginosa* has a wide diversity of virulence factors that are extended from the formation of biofilms, motility, and adhesion to the production of several enzymes, and pyocyanin [3,15,17]. Protease plays a crucial role in the dissociation of the host tissues and establishing the infection [2,20,71]; hence, reducing the production of protease could alleviate bacterial pathogenesis [19,24,25]. Besides extracellular enzymes, *P. aeruginosa* produces bluish-green pigment pyocyanin, which offers protection against oxidative stress inside immune cells and plays a role in *P. aeruginosa* pathogenesis [72,73]. The current findings revealed the significant ability of linagliptin to decrease the production of protease and pyocyanin. Furthermore, an in vivo assessment of the anti-virulence activity of linagliptin was conducted. The liver and kidney tissues were sectioned from mice that were injected with *P. aeruginosa* treated or not with linagliptin. Histopathological examinations revealed immature lesions with mild inflammatory signs in tissues isolated from mice injected with *P. aeruginosa* treated with linagliptin. In contrast, the tissues isolated from mice injected with untreated *P. aeruginosa* showed obvious inflammations, blood vessel congestion, degenerative changes, swelling, and areas of cellular proliferation. Based on the above findings, linagliptin showed a significant ability in vitro to mitigate the *P. aeruginosa* and in vivo to diminish its pathogenesis. 

Quorum sensing (QS) is a chemical language that bacterial cells use to communicate and arrange their virulence [19,74]. The QS system regulates diverse bacterial virulence factors such as the production of virulent enzymes and pigments, bacterial motilities, and biofilm formation [75]. The QS system works in an autoinducer/receptor manner, where the bacterial cells release autoinducers that find their way to the QS receptors on the bacterial cell surfaces. The autoinducer binds to QS receptors to form a complex that will later bind to the virulence genes regulating promotors [74]. Targeting QS systems assures moderation of virulence, and it was assumed as an efficient strategy to mitigate bacterial pathogenesis [14,75]. *P. aeruginosa* mainly utilizes three types of QS systems, two being Lux-types LasI/R and RhlI/R and one a non-Lux-type Pqs, in addition to QscR, which sense the LasI autoinducers [76]. To evaluate the anti-QS activity of linagliptin, quantitative PCR was performed to assay the expression of *P. aeruginosa* QS-encoding genes. Importantly, linagliptin significantly downregulated the encoding genes’ expression of the receptors of the three systems *rhlR, lasR*, and *pqsR* and their synthetases *rhlI*, *lasI*, and *pqsA*, respectively.

The adopted in silico study illustrated the potential LIN affinity towards three *P. aeruginosa* biotargets controlling the virulence genes compared to reported inhibitors. Notably, the residue-wise ligand-target binding interactions were confirmed relevant to several reported small molecules binding towards the same investigated biological targets. For *P. aeruginosa* LasI synthetase, the significance of Arg30 for ligand binding has been described by Gould et al. explaining the Lasi’s substrate specificity in comparison to other AHL synthases, such as EsaI [53]. Deep anchoring of the LIN aromatic scaffold inside the LasI’s elongated tunnel was illustrated owing to the more extended structure of LIN. Generally, deep LIN anchoring into the pocket’s elongated tunnel of *P. aeruginosa* LasI synthase was correlated to the reported better affinity of long-chained acyl AHL at the LasI active site. Additionally, the near co-planner conformation of both the central xanthine ring and terminal quinazoline scaffold at the LIN structure would permit deeper anchoring with minimal steric hindrances towards the LasI’s lining residues within its elongated sub-pocket. The ability of LIN to furnish superior docking binding activity as compared to TZD-C8 reference ligand confers superiority and higher anti-pseudomonal potentiality. Such an inferior docking score could be reasoned to the ligand’s acyl hydrophobic tail simplicity that would furnish lower hydrophobic interactions with lining residues in comparison to the bulky decorated aromatic terminal tail being possessed by LIN.

The LIN’s comparative conformations at *P. aeruginosa* LasI synthase and QscR proteins could be a reason for the differential topology of both binding pockets. Exhibiting a more extended conformation at the QscR’s pocket in regard to the V-shaped one within LasI’s site could be correlated to the earlier larger/more extended pocket size where the big hydrophobic sub-pocket residues require less steric hindrances against LIN anchoring. Unlike the narrow-twisted V-shaped LasI cleft, the QscR showed elongated extended pocket topology, the thing that enables the easy direction of the polar fused rings without significant steric clashes with the pocket’s lining residues. Moreover, the calculated Richard’s SASA/volumes for QscR were of higher values as compared to those of LasI binding sites.

Notably, LIN exhibited more extended polar networks with QscR’s lining amino acids as compared to LasI’s residues. This comparative binding pattern could confer the significant importance of hydrophilic interactions as a major driving force for anchoring small ligands at the QscR pocket. Ligand interactions with Ser38 have been reported as important for determining QscR signal, since this uncharged polar amino acid can preferentially guide the binding of QscR 3-O-HSL over other native unsubstituted compounds [77]. Therefore, significant QscR pocket affinity has been suggested for LIN as depicting relevant hydrogen bonding with the Ser38 sidechain. Similar to polar interactions, the ligand/QscR non-polar interactions were at a higher extent than those for LasI’s pocket, the thing that could be a reason for the preferentially higher overall LIN docking score at QscR binding site. 

The significance of exploring LasR as another *P. aeruginosa* virulence target was that this particular QS protein exhibits promiscuity towards several non-cognate HSL autoinducers produced by several other bacterial species [55]. Thus, the ability of the active LIN to competitively occupy the LasR binding site would deprive *P. aeruginosa* of the opportunity to survive within environments where it can be vastly outnumbered by competing species [78]. The depicted crescent-like conformation for LIN’s terminal aromatic scaffold was a reason for lower LasR binding site SASA/volume as compared to QscR. Despite the LasR’s tight pocket space, LIN successfully accommodated the latter pocket, which was translated into a moderate-to-high negative docking score, while a much superior docking score was assigned for Q9. It was also suggested that the LasR’s pocket promiscuity could be the reason for such a differential binding pose where the non-polar lining residues are of smaller sizes corresponding to those of QscR at the same sequence numbers. Exemplarily, the QscR’s Met127 and Arg132 are corresponding to the LasR’s Ala127 and Val132, respectively. Those latter small-sized amino acids would have imposed lower steric hindered paths against LIN’s terminal aromatic moiety anchoring at LasR’s hydrophobic sub-pocket. Owing to the tightness of the LasR pocket, the large extended LIN conformation could not orient efficiently to secure an optimal hydrogen bond length and angle with Arg61 despite its close proximity towards the LIN’s central xanthine ring (highly decorated with several hydrogen bond acceptors).

Showing comparable hydrophobic contacts for both LIN and Q9 towards several pockets lining residues was suggested as relevant due to the illustrated tightness of the LasI’s pocket, which would bring several non-polar residues at close proximity (≤5 Å) to the LIN’s non-polar groups. Depicting these terminal hydrophobic interactions at the target’s larger sub-pocket was suggested to satisfy the hydrophobic potentiality of the lining residues comprising this end pocket. The latter could be correlated to the depicted high negative scores of these compounds. Based on the above observations, it was clear that Q9 and LIN shared similar hydrophobic binding patterns with LasI’s important non-polar amino acids. Therefore, the differential docked scoring among these docked ligands would be highly correlated to the magnitude and extent of polar bonding interactions being achieved via the ligand’s hydrogen bond acceptor/donor functionalities rather than the hydrophobic ones.

It is worth noting that the above-described LIN-QscR or LasI binding interactions were in good agreement with the current literature. Novel anti-QSTF virtual screened hits of substituted double phenyl rings and a central amide linker showed significant *P. aeruginosa* biofilm inhibition activities [79]. These small molecules exhibited polar interactions with QscR residues Trp62, Tyr66, and/or Asp75, through their amide linkers/sidechains. Further binding stability occurred through π-driven contacts with Tyr66 or Trp90, besides non-polar interactions with Ala41, Tyr52, Val78, Leu82, Ile125, and Met127. The above-described LIN-pocket polar contacts with Try58, Trp62, Asp75, and Ser129 were also found to be significant for stabilizing a series of novel triphenyl- structured antagonists at *P. aeruginosa* QSTF showing higher binding energies than the synthetic triphenyl superinducer [80]. The latter in silico findings were recapitulated by the authors through LasR’s in vitro reporter gene bioassay revealing the significant antagonism of these triphenyl small molecules in presence of native autoinducer, O-C8-HSL [80].

Concerning the ligand-LasI binding, residues Tyr56, Trp60, Asp73, and Ser129 in LasI’s small pocket have been suggested to be important for maintaining the stability of several top-scored hits obtained through virtual screening/pharmacophore-based molecular docking approach of 2373 FDA-approved compounds [81]. The top docked molecule, sulfamerazine, exhibited stability through a molecular dynamic study with favored polar interactions with Tyr56, Trp60, and Ser129 over a 50 ns simulation run. The plant naturally derived flavonoid compound, naringenin, showed time-dependent competition with LasR cognate autoinducer, 3-oxo-C12-HSL, mediating inhibition of *P. aeruginosa* QS regulated genes expression and regulated virulence factors production [82]. The stability of naringenin at LasI binding site was investigated through docking study and polar interactions with Tyr56, Trp60, and Asp73 as well as π-π hydrophobic contacts with Tyr47 sidechains mediated its stability. Finally, seven potential QS inhibitor hits obtained through structure-based high-throughput e-pharmacophore virtual screening out of more than 6000 relevant small molecules also showed extended hydrogen bond network with Trp56, Trp60, Asp73, Thr75, and/or Ser129 for recognized [83].

## 5. Conclusions

Development of bacterial resistance mandates finding new solutions. Mitigating bacterial virulence is an advantageous approach in which safe drugs or compounds can be employed in addition to traditional antibiotics in aggressive bacterial infections. In the current study, linagliptin showed significant in vitro and in vivo anti-virulence and anti-QS activities against *P. aeruginosa*. Furthermore, a detailed molecular docking study attested the considered affinity of linagliptin to QS receptors. The current data revealed clearly the ability of linagliptin and related chemical structures to be repurposed as efficient anti-virulence and anti-QS agents; however, detailed pharmacological and toxicological studies are required prior to clinical use. 

## Figures and Tables

**Figure 1 microorganisms-10-02455-f001:**
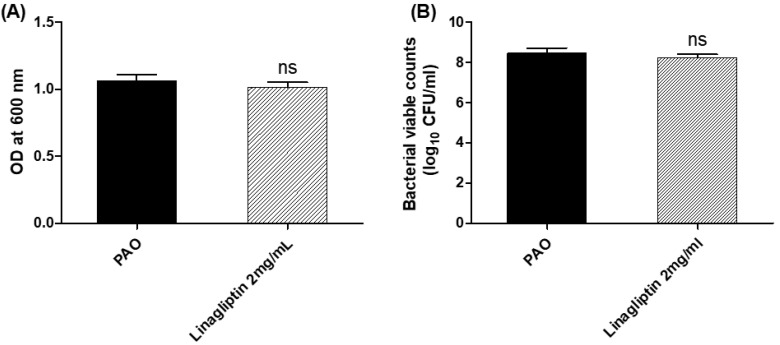
Linagliptin does not affect *P. aeruginosa* growth. To avoid any effect of linagliptin on bacterial growth prior to evaluating its anti-virulence activity; *P. aeruginosa* was grown in the presence or absence of linagliptin at sub-MIC. There was no significant effect of linagliptin at sub-MIC on (**A**) optical densities, and (**B**) bacterial counts of *P. aeruginosa* cultures. ns: non-significant (*p* > 0.05).

**Figure 2 microorganisms-10-02455-f002:**
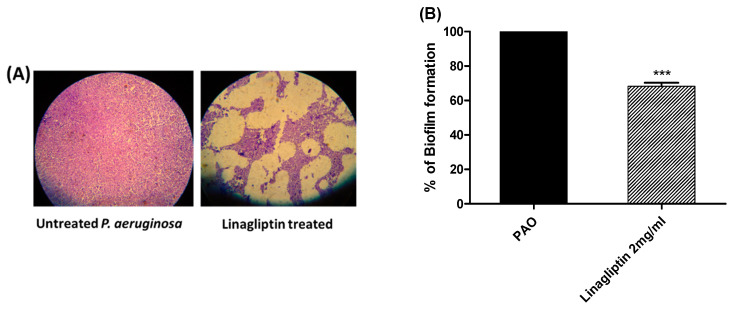
Linagliptin inhibits biofilm formation. The crystal violet method was used to evaluate the anti-biofilm activity of linagliptin at sub-MIC. (**A**) Light microscope images represent the biofilm formation in the presence and absence of linagliptin. Linagliptin markedly decreased the biofilm formation. (**B**) The absorbances of stained biofilm-forming cells were measured in the presence and the absence of linagliptin at sub-MIC. Linagliptin significantly diminished the biofilm formation, *** = *p* ≤ 0.0001.

**Figure 3 microorganisms-10-02455-f003:**
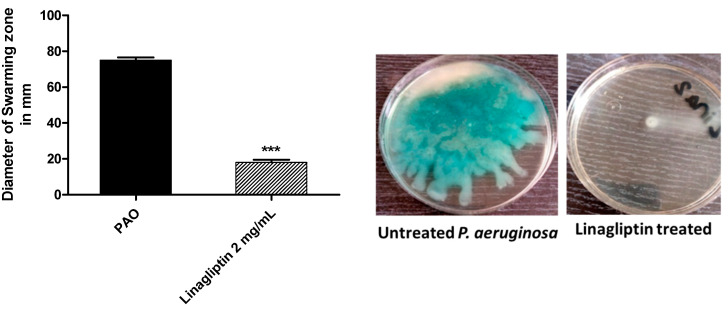
Linagliptin diminishes the *P. aeruginosa* motility. The swarming zones of *P. aeruginosa* on agar plates provided with or without linagliptin at sub-MIC were measured. Linagliptin significantly diminished bacterial motility, *** = *p* ≤ 0.0001.

**Figure 4 microorganisms-10-02455-f004:**
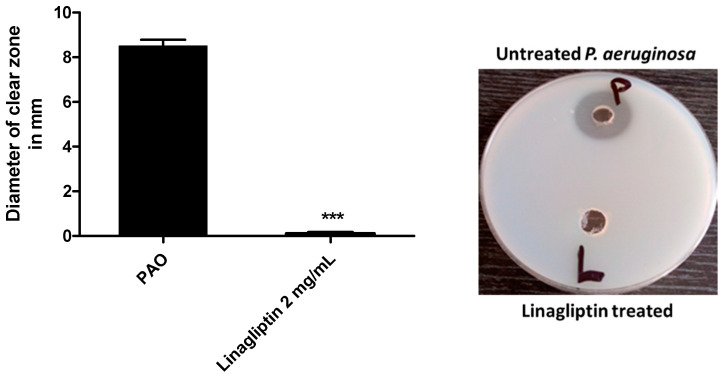
Linagliptin decreases the production of protease by *P. aeruginosa*. The skim milk agar plate method was used to assess the effect of linagliptin on the production of linagliptin. *P. aeruginosa* was grown in the presence or absence of linagliptin at sub-MIC, and the supernatants containing the extracellular protease were collected. Aliquots from supernatants were placed in performed wells and the clear zones were measured after overnight incubation. Linagliptin significantly reduced the production of protease, *** = *p* ≤ 0.0001.

**Figure 5 microorganisms-10-02455-f005:**
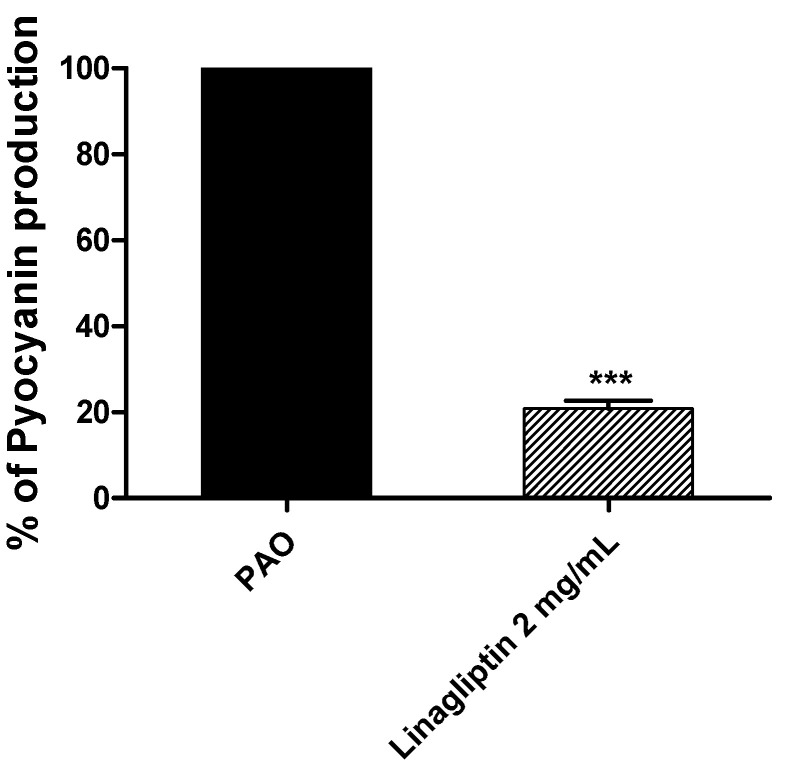
Linagliptin decreases the production of pyocyanin. The absorbance of the produced *P. aeruginosa* pigment was measured in the presence or absence of linagliptin at sub-MIC. Linagliptin significantly reduced the production of pyocyanin, *** = *p* ≤ 0.0001.

**Figure 6 microorganisms-10-02455-f006:**
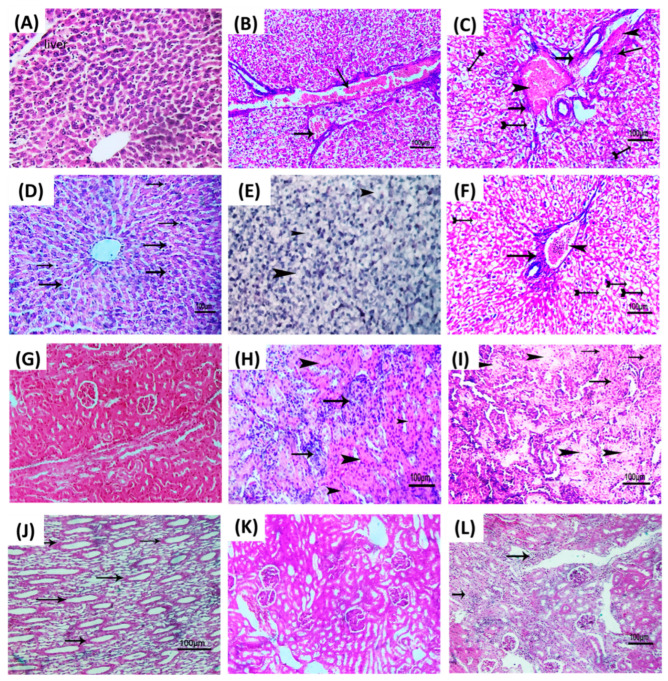
Linagliptin diminishes the *P. aeruginosa* pathogenesis. Histopathological photomicrographs of the liver and kidney tissues (H&E × 200) are captured from mice groups that were uninfected, *P. aeruginosa* infected, or *P. aeruginosa* treated with linagliptin (2 mg/mL). (**A**) Photomicrograph of liver tissues obtained from uninfected (control) mice group showing normal tissue architecture and cellular details. (**B**) Photomicrograph liver section of mice group infected with untreated *P. aeruginosa,* showing severe congestion of hepatic blood vessels (arrows). (**C**) Photomicrograph liver section of mice group infected with untreated *P. aeruginosa* showing perivascular fibrosis (arrows) with congestion of hepatic blood vessel (arrowhead) and hydropic degeneration of some hepatocytes (tailed arrow). (**D**) Photomicrograph liver section of mice group infected with *P. aeruginosa* treated with linagliptin showing mild diffuse infiltration of von Kupffer cells (arrows). (**E**) Photomicrograph liver section of infected mice group infected with *P. aeruginosa* treated with linagliptin showing vacuolation of few hepatocytes in the hepatic parenchyma (arrows). (**F**) Photomicrograph liver section mice group infected with *P. aeruginosa* treated with linagliptin showing mild cellular infiltration (arrow) with congested hepatic blood vessels (arrowhead) and diffuse vacuolation of some hepatocytes (tailed arrows). (**G**) Photomicrograph of kidney tissues obtained from uninfected (control) mice group showing normal renal cortex with normal glomeruli and renal tubules. (**H**) Photomicrograph kidney section of mice group infected with untreated *P. aeruginosa*, showing degenerative changes of renal tubules represented in cloudy swelling (arrowhead) with focal areas of cellular proliferation (arrows). (**I**) Photomicrograph kidney section of mice group infected with untreated *P. aeruginosa,* showing focal caseous necrosis (arrowhead) with cloudy swelling of some renal tubules (arrows) in renal parenchyma. (**J**) Photomicrograph kidney section of mice group infected with *P. aeruginosa* treated with linagliptin showing mild diffuse cystic dilation of some renal tubules (arrows). (**K**) Photomicrograph kidney section of mice group infected with *P. aeruginosa* treated with linagliptin showing apparently normal renal cortex. (**L**) Photomicrograph kidney section of mice group infected with *P. aeruginosa* treated with linagliptin showing fewer focal areas of cellular infiltration (arrows), bar = 100 µm.

**Figure 7 microorganisms-10-02455-f007:**
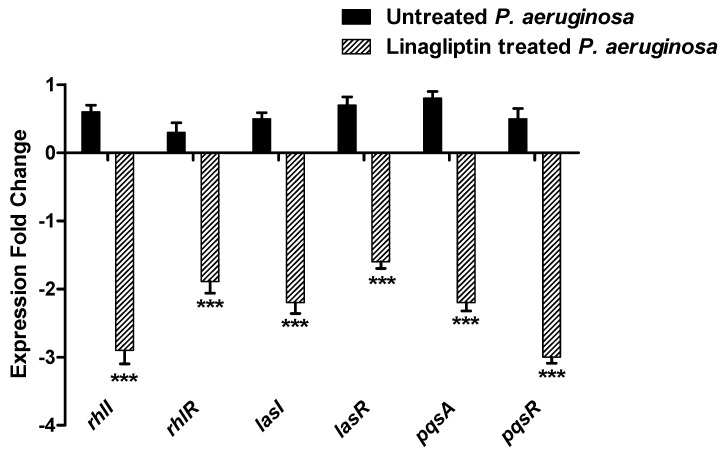
Linagliptin downregulates the expression of *P. aeruginosa* QS−encoding genes. Linagliptin significantly reduced the expression of *P. aeruginosa* LasI/R, RhlI/R and Pqs QS systems. *** = *p* ≤ 0.0001.

**Figure 8 microorganisms-10-02455-f008:**
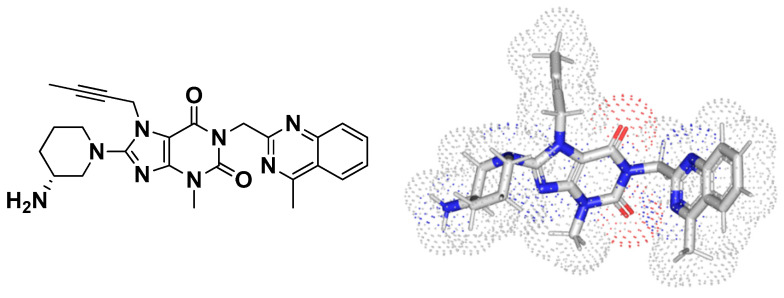
Two- and 3D-dimensional structural representation of linagliptin (LIN).

**Figure 9 microorganisms-10-02455-f009:**
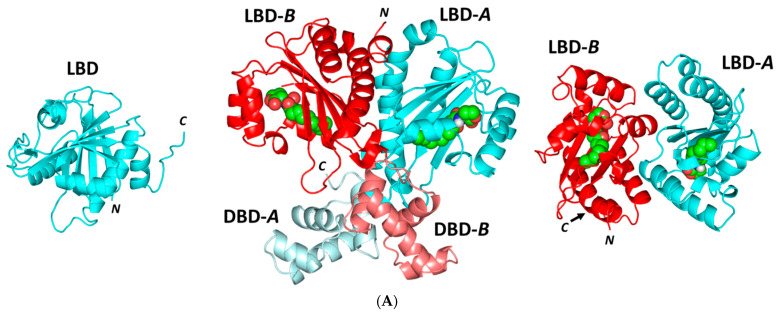
*Pseudomonas aeruginosa* virulence-regulating biological targets. (**A**) Overall cartoon representations of the ligand-binding domain (LBD) concerning LasI-type synthase (right); QscR (middle), and LasR (left) QS targets. Each QS protein is differentially colored based on LBD and/or DNA-binding domain (DBD) as dark/light cyan or red for protomer-A or -B, respectively. Ligands are green spheres. (**B**) Calculated putative pockets of the *P. aeruginosa* biological targets via CASTp server (1.4 Å radius probes) colored in different colors (blue and red) for each target protomer. (**C**) Surface representation of binding sites with an overlay of co-crystallized ligand (green sticks) of its respective bacterial target, C12-HSL and 3-O-C10-HSL. Hydrogen bonds are red dashed lines and residues (cyan lines) located within 4 Å radius of the bound ligand, and are displayed and labeled with a sequence number.

**Figure 10 microorganisms-10-02455-f010:**
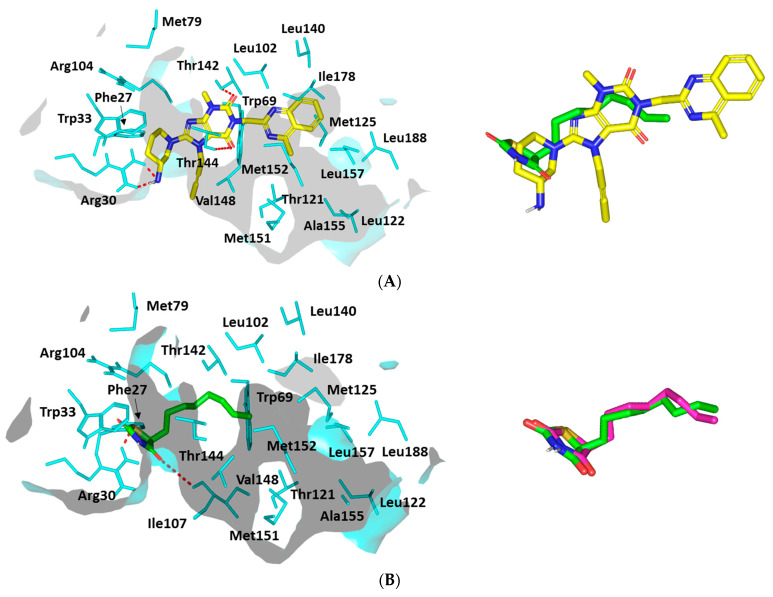
Ligand-protein binding interactions at *Pseudomonas aeruginosa* LasI synthase binding site. Predicted binding modes of the docked ligands (**A**) LIN and (**B**) TZD-C8 as cyan and green sticks, respectively. Only residues located within 5 Å radius of bound ligands are displayed (cyan lines) and labeled with a sequence number. Non-polar hydrogens are removed for clarity and hydrogen bonds are depicted as red dashed lines. The right panels are an overlay of LIN and docked reference standard, TZD-C8, as well as, the docked reference ligand and its re-docked state (magenta sticks).

**Figure 11 microorganisms-10-02455-f011:**
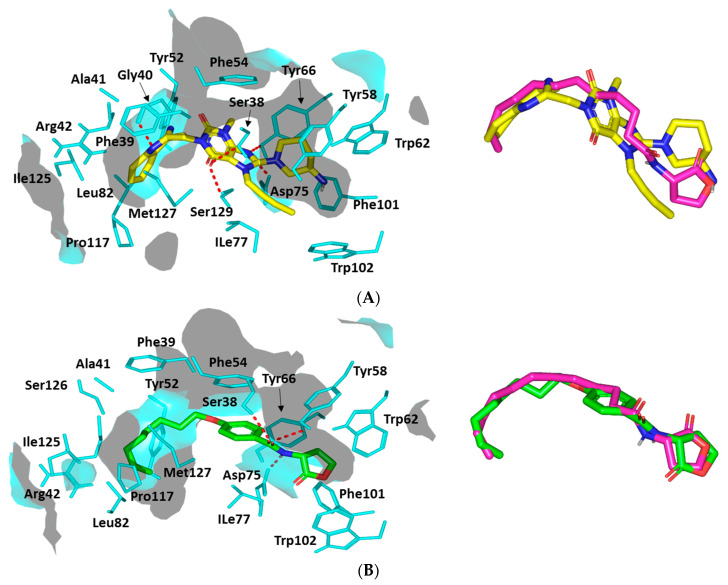
Ligand–protein binding interactions at *Pseudomonas aeruginosa* QscR quorum-sensing transcription protein binding site. Predicted binding modes of the docked ligands (**A**) LIN and (**B**) Q9 as cyan and green sticks, respectively. Only residues located within 5 Å radius of bound ligands are displayed (cyan lines) and labeled with a sequence number. Non-polar hydrogens are removed for clarity and hydrogen bonds are depicted as red dashed lines. The right panels are an overlay of LIN or Q9 with the co-crystalline ligand, C12-HSL, as magenta sticks.

**Figure 12 microorganisms-10-02455-f012:**
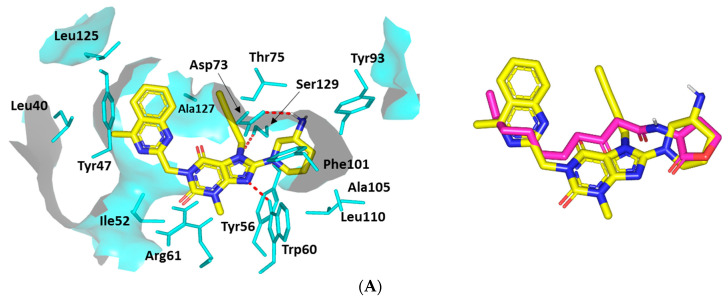
Ligand-protein binding interactions at *Pseudomonas aeruginosa* LasR quorum-sensing transcription protein binding site. Predicted binding modes of the docked ligands (**A**) LIN and (**B**) Q9 as cyan and green sticks, respectively. Only residues located within 5 Å radius of bound ligands are displayed (cyan lines) and labeled with a sequence number. Non-polar hydrogens are removed for clarity and hydrogen bonds are depicted as red dashed lines. The right panels are an overlay of LIN or Q9 with the co-crystalline ligand, 3-O-C10-HSL, as magenta sticks.

**Table 1 microorganisms-10-02455-t001:** Sequences of the primers used in this study.

Target Gene	Sequence (5′–3′)	Reference
*lasI*	For: CTACAGCCTGCAGAACGACA Rev: ATCTGGGTCTTGGCATTGAG	[2,21,22]
*lasR*	For: ACGCTCAAGTGGAAAATTGG Rev: GTAGATGGACGGTTCCCAGA	[2,21,22]
*rhlI*	For: CTCTCTGAATCGCTGGAAGG Rev: GACGTCCTTGAGCAGGTAGG	[2,3,21,22]
*rhlR*	For: AGGAATGACGGAGGCTTTTT Rev: CCCGTAGTTCTGCATCTGGT	[2,21,22]
*pqsA*	For: TTCTGTTCCGCCTCGATTTC Rev: AGTCGTTCAACGCCAGCAC	[2,21,22]
*pqsR*	For: AACCTGGAAATCGACCTGTG Rev: TGAAATCGTCGAGCAGTACG	[2,21,22]
*rpoD*	For: GGGCGAAGAAGGAAATGGTC Rev: CAGGTGGCGTAGGTGGAGAAC	[2,21,22]

## Data Availability

Not applicable.

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
