# Peer review of "Hiring of the Anti-Quorum Sensing Activities of Hypoglycemic Agent Linagliptin to Alleviate the Pseudomonas aeruginosa Pathogenesis"

_microorganisms, 2022, doi:10.3390/microorganisms10122455_

Round 1
Reviewer 1 Report
In this manuscript, the authors comprehensively research that hiring of the anti-quorum sensing activities of hypoglycemic agent linagliptin. And explain the anti-quorum sensing mechanism of linagliptin by molecular docking. However, the content of the manuscript needs to be improved.
Details on suggestions and comments are given below:
1. In line 116 on page 3,authors used Linagliptin to evaluate the effect of P. aeruginosa growth, and chose sub-MIC (1/5 MIC) to evaluate. However, the author did not mention why this concentration was selected for evaluation.
2. In line 169 on page 5, authors were arranged three-weeks old Mus musculus mice to five groups, but only four groups are mentioned below. And treated P. aeruginosa with DMSO, whether to set DMSO control group?
3. In line 344 on page 11, authors described LasI protein in detail and illustrated it with pictures, but did not describe other proteins in detail below. It shall be illustrated accordingly.
4. In the whole article, only one P. aeruginosa is mentioned as the experimental group. Whether it can cover all situations is a question that the authors need to think about.
5. Materials and Methods 2.3:The method used for bacterial cell counting was not described in the text.
6. Materials and Methods 2.4:Generally, the bacterial biofilm formation process is divided into five stages: (i) initial/reversible attachment, (ii) irreversible attachment, (iii) formation of microcolonies, (iv) maturation, and (v) cellular detachment/dispersion. The author chose to cultivate for 24 hours of biofilm. What period was it, and why did he choose this period?
7. Materials and Methods 2.9:In this experiment, a total of 25 mice were killed by cervical dislocation. Is there any ethical clearance? Not found in the text.
Minor comments
1. In line 709 on page 22, “P. aeruginosa” should be italicized.
2. In line 711 on page 22, “Pseudomonas aeruginosa” should be abbreviated.
3. In line 712 on page 22, “Pseudomonas aeruginosa” should be abbreviated.
4. Result 3.6:Figure 6:The bar of some figures were not marked, so it is recommended that they could be clearly marked.
5. Line 342,343,373,418:The serial numbers of these titles are incorrectly marked.
Author Response
Dear Reviewer,
Thank you very much for your valuable and constructive comments and suggestions. Please find the attached reply to all the raised points.
Best regards,
Wael

Reviewer 2 Report
The manuscript entitle: "Hiring of the anti-quorum sensing activities of hypoglycemic 2 agent linagliptin to alleviate the Pseudomonas aeruginosa path- 3 ogenesis" by Khayat et al. describes the anti-quorum sensing activity of known anti-diabetic compound compound "linagliptin". Though the manuscript is well-written and several experiments were carried out to establish anti QS activity of linagliptin. However, the most important concern associated with this work is the very high concentration of linagliptin (2mg/mL) used to evaluate the activity. Authors should perform these experiments at lower concentrations (micomolar).
Also, in silico molecular docking study was carried out only on the LasR receptor. Can author/s explain why not the PqsR receptor?
Other minor corrections:
Abstract: In-vitro, In-vivo, and In-silico must be in italics
References: The names of PA and other bacteria must be in italics throughout.
Author Response
Dear Reviewer,
We are very thankful for your valuable comments and suggestions. Please find the attached reply to all the raised points.
Best Regards,
Wael

Round 2
Reviewer 2 Report
The manuscript has been revised and improved significantly and can be published in Microorganisms in its current form.